# Endoscopic Contrast-Enhanced Ultrasound and Fine-Needle Aspiration or Biopsy for the Diagnosis of Pancreatic Solid Lesions: A Systematic Review and Meta-Analysis

**DOI:** 10.3390/cancers16091658

**Published:** 2024-04-25

**Authors:** Giorgio Esposto, Giuseppe Massimiani, Linda Galasso, Paolo Santini, Raffaele Borriello, Irene Mignini, Maria Elena Ainora, Alberto Nicoletti, Lorenzo Zileri Dal Verme, Antonio Gasbarrini, Sergio Alfieri, Giuseppe Quero, Maria Assunta Zocco

**Affiliations:** 1CEMAD Digestive Disease Center, Fondazione Policlinico Universitario “A. Gemelli” IRCCS, Università Cattolica del Sacro Cuore, 00168 Rome, Italy; giorgio.esposto@guest.policlinicogemelli.it (G.E.); linda.galasso@guest.policlinicogemelli.it (L.G.); raffaele.borriello01@icatt.it (R.B.); irene.mignini@guest.policlinicogemelli.it (I.M.); mariaelena.ainora@policlinicogemelli.it (M.E.A.); antonio.gasbarrini@unicatt.it (A.G.); 2Digestive Surgery Unit, Fondazione Policlinico Universitario “A. Gemelli” IRCCS, Università Cattolica del Sacro Cuore, 00168 Rome, Italy; giuseppe.massimiani01@icatt.it (G.M.); sergio.alfieri@unicatt.it (S.A.); giuseppe.quero@unicatt.it (G.Q.); 3Internal Medicine and Gastroenterology Unit, Fondazione Policlinico Universitario “A. Gemelli” IRCCS, Università Cattolica del Sacro Cuore, 00168 Rome, Italy; paolo.santini@guest.policlinicogemelli.it (P.S.); alberto.nicoletti@policlinicogemelli.it (A.N.); lorenzo.zileridalverme@policlinicogemelli.it (L.Z.D.V.)

**Keywords:** solid pancreatic lesions, endoscopic contrast-enhanced ultrasound, fine-needle aspiration

## Abstract

**Simple Summary:**

We conducted a systematic search of the literature to explore if endoscopic contrast-enhanced ultrasound (ECEUS) could improve the diagnostic success of pancreatic solid lesion biopsy or fine needle aspiration. The analysis that we conducted on 1.178 patients showed a slight trend of more diagnoses and the greater efficacy of a single pass in patients who underwent contrast-guided pancreatic sampling, although this finding did not reach statistical significance. We believe that our analysis provides a useful insight for clinical practice and could aid future investigations on this topic.

**Abstract:**

Introduction: Endoscopic ultrasound-guided fine-needle aspiration (EUS-FNA) and endoscopic ultrasound-guided fine-needle biopsy (EUS-FNB) are currently recommended for the pathologic diagnosis of pancreatic solid lesions (PSLs). The application of contrast-enhanced endoscopic ultrasound (ECEUS) could aid the endoscopist during an FNA and/or FNB procedure. CEUS is indeed able to better differentiate the pathologic tissue from the surrounding healthy pancreatic parenchyma and to detect necrotic areas and vessels. Objectives: Our objective was to evaluate if ECEUS could reduce the number of needle passes and side effects and increase the diagnostic efficacy of FNA and/or FNB. Methods: A comprehensive literature search of clinical studies was performed to explore if ECEUS-FNA or FNB could increase diagnostic accuracy and reduce the number of needle passes and adverse effects compared to standard EUS-FNA or FNB. In accordance with the study protocol, a qualitative and quantitative analysis of the evidence was planned. Results: The proportion of established diagnoses of ECEUS was 90.9% compared to 88.3% of EUS, with no statistically significant difference (*p* = 0.14). The diagnosis was made through a single step in 70.9% of ECEUS patients and in 65.3% of EUS patients, without statistical significance (*p* = 0.24). The incidence of adverse reactions was substantially comparable across both groups (*p* = 0.89). Conclusion: ECEUS-FNA and FNB do not appear superior to standard EUS-FNA and FNB for the diagnosis of pancreatic lesions.

## 1. Introduction

The current rate of pancreatic cancer (PC) misdiagnosis reaches values varying from 14% up to 35% in the case of mass-forming chronic pancreatitis [1,2]. This may be due to the increased PC incidence rate as well as the absence of adequate preoperative workups, especially in the case of no clear radiological features of pancreatic solid lesions (PCLs). Thus, the need for an extensive pretreatment diagnostic assessment of PSLs in order to avoid mistakes that could impair patients’ quality of life and overall survival is implicit. Endoscopic ultrasound-guided fine-needle aspiration (EUS-FNA) and endoscopic ultrasound-guided fine-needle biopsy (EUS-FNB) are currently recommended for the pathologic diagnosis of PSLs [3,4]. Although both these sampling techniques demonstrate specificity of up to 100%, current sensitivity ranges from 82% to 90.4% for EUS-FNA and from 90% to 100% for EUS-FNB, while the diagnostic accuracy ranges from 78% to 90% for EUS-FNA and from 87% to 98% for EUS-FNB [5,6,7,8]. Moreover, in several cases, multiple needle passes are needed to guarantee a sufficient sample for pathological analysis, possibly increasing the risk of tumor seeding and other procedural-related side effects [9,10]. Several factors were explored in order to increase the diagnostic performance of EUS-FNA and EUS-FNB, such as the type and caliper of the needle, the sampling technique, the rapid on-site cytopathology (ROSE) and the more recent application of novel techniques such as endomicroscopy, elastography and, above all, contrast-enhanced endoscopic ultrasound (ECEUS). Regarding this latter technique, several advantages need to be highlighted as compared to other diagnostic procedures. For instance, CEUS is able to better differentiate pathologic tissue from the surrounding healthy pancreatic parenchyma and can detect necrotic areas and vessels, which allows PSLs to be targeted more effectively, reducing, at the same time, potential side effects such as postprocedural bleeding [11]. A meta-analysis has demonstrated a statistically significant advantage of ECEUS-FNA over standard EUS-FNA, especially in terms of diagnostic accuracy and sensitivity for lesions greater than 2 cm in diameter [12]. However, ECEUS-FNA was not superior in terms of the reduced number of needle passes and the procurement of an adequate histologic core. Moreover, this meta-analysis did not take into account any comparison between FNB and FNA techniques either regarding the diagnostic outcomes and the potential side effects. Based on these premises, the aim of this systematic review is to evaluate the current evidence on the application of CEUS for pancreatic tissue sampling, in order to report the state of the art and give a comprehensive overview on the sensitivity, NPV, number of needle passes and side effects either with FNA or FNB.

## 2. Materials and Methods

### 2.1. Research Question

A systematic literature review was conducted to answer the following research question: “Can ECEUS increase the diagnostic success of pancreatic tissue sampling and reduce the number of needle passes and side effects of the procedure?”

### 2.2. Protocol Registration

The study was conducted in conformity with the preferred reporting items for systematic reviews and meta-analyses (PRISMA) [13,14] and synthetized with a meta-analysis. The study protocol for this systematic review and meta-analysis was written and submitted to the International Register of Systematic Reviews (PROSPERO, ID: CRD42024512614), prior to the completion of the literature search.

### 2.3. Literature Search Strategy

The search was conducted in the following electronic bibliographic databases: Medline (via PubMed), Embase (via Ovid) and Web of Science. To guarantee adequate sensitivity, we designed the search strategy to only include terms related to the diagnostic technique being evaluated and the target population of patients affected by pancreatic cancer. Therefore, three domains were combined regarding endoscopic contrast-enhanced ultrasound and pancreatic cancer. The search string for each database can be seen in the Appendix A. The search was completed by manually reviewing references of retrieved articles and the prior systematic reviews or meta-analyses on this topic.

### 2.4. Selection Criteria

Studies were considered eligible if they met the following inclusion criteria: (1) randomized trial; (2) non-randomized studies of interventions; (3) written in English; (4) patients with pancreatic lesions who underwent ECEUS-guided fine-needle aspiration (FNA) or biopsy for diagnosis. Meeting abstracts and oral or poster presentations given at scientific congresses were excluded. The results of the literature search were merged, and duplicates were removed using EndNote^TM^. Individual records were manually screened with title and abstract analysis by three independent reviewers (G.E., G.M. and L.G.). Any disagreement was resolved by discussion. Records considered appropriate were eligible for full-text analysis. Study selection, full-text analysis and data extraction were performed by three reviewers (G.E., G.M. and L.G.). In the case of more than one record reporting on a single study, we focused on the most recently published paper in which the outcomes of the review were reported in the most exhaustive and complete way.

### 2.5. Data Extraction and Data Synthesis

The following data were collected: author, location, year of publication, study design, total number of patients, histology of pancreatic cancer, DCEUS continuous parameters derived from time–intensity curves, type of contrast agent used, software used for the analysis of time–intensity curves and confounding factors as reported in each study. Missing data were requested from the studies’ authors. In accordance with the study protocol, a qualitative analysis of the evidence was conducted. The results were summarized in a comprehensive summary table of study characteristics and baseline characteristics of participant patients (Table 1 and Table 2). Data synthesis was carried out by dividing the selected studies into groups defined by endoscopic technique (i.e., ECEUS vs. standard EUS). The results were summarized with a comprehensive summary table of the outcomes (Table 3). Through quantitative analysis, it was possible to calculate the proportion of established diagnoses, the cumulative incidence of adverse reactions and the proportion of established diagnoses obtained through a single step in the two groups of patients who underwent CEUS and standard EUS. For the proportion of established diagnoses, it was possible to include eight studies [15,16,17,18,19,20,21,22].

For the proportion of adverse events, it was possible to consider seven works [15,16,18,19,20,21,23]. For the proportion of established diagnoses obtained through a single step, it was possible to include three works [15,16,18]. The distribution of the aforementioned outcomes was then analyzed using the χ² test to verify the presence of any statistically significant differences between the two groups.

### 2.6. Risk-of-Bias Assessment

The risk of bias of the eligible studies was assessed using the revised Cochrane risk-of-bias tool for randomized trials (RoB-2 tool version 2) and the Risk of Bias In Non-Randomized Studies of Interventions (ROBINS-I) tool (version 1) [24,25,26]. Risk-of-bias assessment was carried out by three authors (G.E., G.M. and L.G.), and any disagreement between the two independent reviewers was resolved by discussion, with the involvement of a fourth review author where necessary. The number of studies was too small to allow a graphical assessment of publication bias by funnel plot or statistical assessment by Egger’s test.

### 2.7. Outcomes

The main outcome of the current systematic review was the evaluation of the diagnostic success of ECEUS-guided tissue acquisition intended as the capability to obtain the correct histopathological diagnosis of pancreatic mass compared to standard EUS-guided tissue acquisition. The gold standard for diagnosis is considered either surgical histopathology for patients candidates for surgery or unequivocally malignant histopathology with clinical and radiological findings compatible with the diagnosis. A benign histological exam by FNA or FNB must be confirmed by surgical pathology or by a follow-up of at least 6 months. Secondary outcomes were the proportion of diagnosis at first pass and the occurrence of periprocedural adverse events (i.e., bleedings and/or acute pancreatitis) between ECEUS-FNA and EUS-FNA. It was not possible to analyze the mean number of passes required to obtain a diagnosis since it was not reported by all studies. Therefore, considering the available data, we considered the proportion of diagnoses at first pass as a secondary outcome.

## 3. Results

### 3.1. Study Selection

Three biomedical databases were screened using the prespecified search methods on 20 December 2023, and a total of 1945 studies were found (Medline via PubMed: 1630; Embase via Ovid: 181; and Web of Science: 134). After the removal of duplicates, 1784 records underwent primary eligibility screening based on titles and abstracts. As a result, 315 papers met the eligibility criteria for full-text analysis. A total of 308 studies were excluded: 295 papers had different outcomes, 2 were only presented as abstracts, 1 paper had incomplete full text and 4 studies had a different study design. Five narrative reviews and one previously published meta-analysis were excluded from further analysis. After a thorough analysis of the references of each paper, we found one article in the meta-analysis bibliography that matched our review question; therefore, one study was added to the list of eligible studies. Finally, nine studies matched the predetermined eligibility requirements for this systematic review. After a structured risk-of-bias assessment, seven original papers were included in the quantitative analysis. Figure 1 shows the PRISMA selection flow diagram that describes the study-selection process in detail.

### 3.2. Risk-of-Bias Assessment

To evaluate the internal and external validity of each included study, a structured analysis of the risk of bias was carried out using the revised Cochrane risk-of-bias tool for randomized trials (RoB 2 tool) [24,25] and the Risk of Bias in Non-Randomized Studies of Interventions (ROBINS-I) tool [26] (Figure 2 and Figure 3). It is important to clarify that the risk-of-bias assessment evaluated each included study in the context of the research questions of the current systematic review and meta-analysis and did not analyze the general scientific worth or quality of each individual study.

Two randomized studies displayed some causes for concern in the overall risk of bias according to the RoB2 tool [15,16]. The study by Sugimoto et al., 2015 [15] was of some concern with regard to the “Randomizing Process” domain, because it was not clear if the allocation sequence was concealed. The studies by Sugimoto et al., 2015 [15] and Cho et al., 2021 [16] were of some concern with regard to the “Deviation from the intended intervention” domain, because it is not specified whether analyses to estimate the effect of assignment were conducted or not. The remaining two randomized studies [17,18] had a low overall risk of bias according to the RoB2 tool. Three non-randomized interventional trials had a low overall risk of bias according to the ROBINS-I tool [19,21,22]. The studies by Itonaga et al. [23] and Seicean et al., 2015 [20] showed a serious risk of bias in the “Confounding” domain due to the study designs themselves. Indeed, both studies were designed as crossover studies, possibly leading to bias in the conduction of ECEUS-FNA after EUS-FNA. This bias is analog to the carry-over bias of crossover clinical trials. As the application of a therapeutical intervention could have residual effects on the response to a second therapy, the results of EUS-FNA could bias the conduction of ECEUS-FNA on the same lesion.

### 3.3. ECEUS vs. Standard FNA or Biopsy of Pancreatic Solid Lesions: Qualitative Summary

Nine studies were considered eligible for qualitative synthesis: four randomized studies and five non-randomized studies of intervention. All studies were homogenous in terms of procedure, but there was a certain measure of heterogenicity in outcomes and pancreatic lesion type. Not every study explored the optimal number of needle passes, the presence of periprocedural adverse events or the adequacy of the specimen. 

The study characteristics of the included publications are displayed in Table 1, while the baseline characteristics of patients included in each study are summarized in Table 2. The number of study participants ranges from 40 to 240. The details of the outcomes are summarized in Table 3 and Table 4. 

Sugimoto et al. [15] conducted a prospective randomized study involving 40 patients with solid pancreatic lesions (SPLs). Patients were allocated either to the ECEUS-FNA group (n = 20) or to the standard EUS FNA group (n = 20). The main outcome was the number of final diagnoses obtained by histological exam. Among ECEUS -FNA patients, 19 out of 20 were diagnosed with pancreatic cancer, and in the conventional EUS-FNA group, 19 out of 20 were diagnosed with pancreatic cancer. With regard to the number of needle passes, ECEUS-FNA demonstrated a significantly higher rate (60%) of obtaining sufficient biopsy samples with only one needle pass compared to conventional EUS-FNA (25%) (*p* = 0.03). Both groups achieved a 100% sampling rate, with comparable sensitivity and accuracy (90% vs. 85%). No complications or evidence of tumor seeding were observed in either group throughout the clinical course of operable and inoperable cases. The results suggest that ECEUS-FNA could potentially reduce the number of required needle passes, offering a valuable and safe diagnostic technique for SPLs. 

Cho et al. [16] designed a multicenter study and prospectively enrolled 240 patients with SPLs and randomly allocated them to the ECEUS-FNA group (n = 120) or to the EUS-FNA group (n = 120). The main outcome was the comparison of diagnostic sensitivity and the determination of the optimal number of needle passes for accurate pathological diagnosis. Despite random assignment, the baseline characteristics between the ECEUS and conventional EUS groups showed no significant differences. The diagnostic sensitivity of ECEUS-guided FNA/FNB sampling was comparable to conventional EUS (86.8% vs. 91.1%). Interestingly, the first needle pass diagnosed 68.3% of cases, with the optimal number of needle passes identified as three. Unexpected adverse events, including postprocedural pancreatitis, occurred in six cases, which were evenly distributed between the two groups. ECEUS did not significantly enhance the diagnostic efficacy or reduce the number of required needle passes compared to conventional EUS. Univariate logistic regression identified larger needle diameters and FNB needles as factors that increase diagnostic sensitivity. Nevertheless, in multivariate analysis, no independent factors improving diagnostic sensitivity were identified. The study concluded that, despite limitations such as variations in FNA/FNB sampling protocols and the absence of on-site pathologists, ECEUS did not consistently outperform conventional EUS in diagnosing SPLs.

In 2020, Seicean et al. [17] enrolled 150 patients with solid pancreatic masses in a prospective randomized study. After an initial EUS evaluation, patients were randomly allocated to EUS-FNA or ECEUS-FNA groups (75 patients for each arm). The main outcome was the determination of whether ECEUS-FNA offered superior diagnostic accuracy. The final analysis included 148 patients (2 patients were lost at follow-up), with adenocarcinoma being the most common diagnosis (66.9%). Both ECEUS-FNA and EUS-FNA demonstrated similar diagnostic performances without any significant differences. Hypoenhancement in ECEUS was particularly accurate for adenocarcinoma, while isoenhancement and hyperenhancement were valuable for non-adenocarcinoma tumors. This study indicated no significant differences in diagnostic accuracy between the two methods, challenging the expectation that ECEUS-FNA would outperform standard EUS-FNA.

Kuo et al. [18] enrolled 120 patients in a prospective randomized controlled study. Patients were divided into ECEUS FNB (n = 60) or EUS-FNB with fanning technique (n = 60) groups. Two patients were excluded after EUS evaluation: one with a pancreatic cystic lesion in the ECEUS group and one with an ampullary tumor in the fanning group. The main outcome was the determination of if ECEUS could potentially reduce the number of required needle passes, while the secondary outcomes were diagnostic accuracy and adverse events. The authors found no significant differences in the number of needle passes between the two groups. The diagnostic accuracy of first and second needle passes was higher in the ECEUS group, although it did not reach statistical significance (first pass 76.3% vs. 72.9%, *p* = 0.83; second pass 91.5% vs. 86.4%, *p* = 0.56). Regarding the secondary outcomes, there was no statistical difference between the contrast group (sensitivity 100% and specificity 66.7%, *p* = 1) and fanning group (sensitivity 100% and specificity 100%, *p* = 1). Diagnostic accuracy was similar in the two groups (98.3% vs. 100%, *p* = 1). No difference was found among adverse events (*p* = 0.68).

Hou et al. [19] included 163 patients in a retrospective case–control study. The two cohorts studied were patients who underwent ECEUS-guided FNA or standard EUS-guided FNA on solid pancreatic lesions. ECEUS diagnoses were independently reviewed, and the aspiration site was selected based on lesion characteristics. In the EUS group, color Doppler was used to exclude vessels before sampling. Statistical analyses, including receiver operating characteristic (ROC) analyses, were conducted on 163 eligible patients, revealing comparable performance between ECEUS-FNA and EUS-FNA in diagnosing malignant lesions. ECEUS-FNA was slightly more expensive but demonstrated cost-effectiveness per diagnostic sample (ECEUS-FNA, AUC = 0.91, sensitivity = 81.6%, specificity = 100%, positive predictive value = 100%, negative predictive value = 74.1% and accuracy = 87.9%). Moreover, ECEUS allowed a greater number of adequate biopsy specimens compared to standard EUS (96.6% vs. 86.7%). The authors concluded that simultaneous ECEUS during FNA improves the diagnostic yield and sampling of solid pancreatic lesions, proving to be a valuable tool for their evaluation. However, the study’s limitations, such as its retrospective nature, emphasize the importance of large prospective studies to confirm these findings. 

Itonaga et al. [23] prospectively enrolled 93 consecutive patients with SPLs. These patients underwent EUS-FNA; the first needle pass was conducted via standard EUS, while the second needle pass was conducted via ECEUS. The samples obtained were then compared in order to test the accuracy rate of diagnosis and the percentage of appropriate sampling for histological examination. Sensitivity and percentage of appropriate sampling were statistically higher among ECEUS-FNA than EUS-FNA (84.9% vs. 68.8%, *p* = 0.003 and 76.5% vs. 58.8%, *p* = 0.01, respectively). This difference was particularly evident in the head and body/tail of the pancreas and when the mass size exceeded 15 mm. ECEUS-FNA also exhibited higher rates in the non-enhancing and homogeneous groups. 

In 2015, Seicean et al. [20] conducted a prospective study on 58 patients with SPLs. After exclusions and follow-up losses, 51 patients completed the study. Each patient underwent FNA both via standard EUS and ECEUS. In total, 95.12% of malignant tumors were classified as hypoenhanced on ECEUS. ECEUS-FNA showed greater sensitivity, negative predictive value and accuracy (9.7%, 9% and 8% greater) compared to standard EUS-FNA, although these differences were not significant. The length of core samples was insignificantly lower for EUS-FNA than ECEUS-FNA. The combination of both methods significantly improved diagnostic accuracy (*p* = 0.03). In conclusion, the study demonstrated that ECEUS-FNA significantly enhances diagnostic accuracy over EUS-FNA alone for solid pancreatic masses. The combined approach showcases high diagnostic accuracy and a notable negative predictive value for malignancy, affirming the feasibility and safety of ECEUS-FNA in clinical practice. 

Facciorusso et al. [21] prospectively enrolled 362 patients with pancreatic lesions who were candidates for EUS-FNA (n = 250) or ECEUS-FNA (n = 112). The main outcome was diagnostic accuracy, while the secondary outcomes were sample adequacy, the number of needle passes required for an adequate sample and the number of adverse effects. Patients were then matched by one-to-one propensity-score matching into two groups of 103 patients each. There were no adverse events. The diagnostic sensitivity of the ECEUS group was 87.6%, while the EUS group displayed a sensitivity of 80% (*p* = 0.18). The negative predictive value was 56% in the ECEUS group and 41.5% in the EUS group (*p* = 0.06). Diagnostic accuracy was comparable across both groups (ECEUS 89.3% vs. EUS 82.5%) (*p* = 0.40), as was the case for sample adequacy (ECEUS 94.1% vs. EUS 91.2%) (*p* = 0.42). No statistical differences were seen in the rate of sample adequacy (ECEUS 33% vs. EUS 28.1%, *p* = 0.44) and in the optimal number of needle passes (ECEUS 2.4 ± 0.6 vs. EUS 2.7 ± 0.8, *p* = 0.76). Overall, ECEUS-FNA did not showcase statistical superiority compared to standard EUS-FNA. 

Lai et al. [22] retrospectively collected data from 155 patients with pancreatic lesions or retroperitoneal lymph nodes who underwent EUS-FNB or ECEUS-FNB. The outcomes were diagnostic accuracy and the optimal number of needle passes to obtain a histological diagnosis. A total of 133 out of 155 patients were diagnosed with cancer, and these 133 patients were divided into the EUS-FNB group (n = 85) or the ECEUS-FNB group (n = 48). The overall number of needle passes was statistically lower in the ECEUS group (2.21 ± 0.68) than in the EUS group (3.64 ± 1.2) (*p* < 0.001). The subanalysis of needle passes for tumor location confirmed these results: ECEUS 2.17 ± 0.76 vs. EUS 3.64 ± 1.04 for uncinate and head lesions (*p* < 0.001) and ECEUS 2.45 ± 10.69 vs. EUS 3.67 ± 1.39 for body and tail lesions (*p* = 0.02). There was no difference in the diagnostic accuracy between the two groups, although a trend toward a higher success rate of histological diagnosis was seen in the ECEUS group.

### 3.4. ECEUS vs. Standard FNA or Biopsy of Pancreatic Solid Lesions: Meta-Analysis

The proportion of established diagnoses in patients who underwent ECEUS was 90.9%, compared to 88.3% in those subjected to standard EUS-FNA and/or FNB, with no statistically significant differences between the two groups (*p* = 0.14) (Table 5). 

The diagnosis was made through a single step in 70.9% of patients who underwent CEUS and in 65.3% of patients who underwent standard EUS-FNA and/or FNB, without statistically significant differences (*p* = 0.24) (Table 6). The incidence of adverse reactions was substantially comparable between the two groups (1.0% vs. 0.9%, with and without CEUS, respectively) (*p* = 0.89) (Table 7).

Two sensitivity analyses in patients undergoing FNA and FNB showed similar results to the main analysis. Patients undergoing FNA received an established diagnosis in 85.7% of patients undergoing EUS and in 89.4% of patients with ECEUS, with a slight trend of greater diagnosis in patients who underwent the contrastographic method but in the absence of a statistically significant difference in the χ² test (*p* = 0.09). Similarly, in patients undergoing FNB, no difference was noted in the proportion of established diagnoses between those who underwent ECEUS or not (98.0 vs. 95.5%, *p* = 0.31). The role of the covariate relating to the sampling method (i.e., FNA vs. FNB) was evaluated using the homogeneity test, which showed a *p* = 0.62 with evidence of homogeneity between the FNA and FNB strata. The estimate of the Relative Risk (RR) of established diagnosis among those who underwent ECEUS and EUS in the two FNA and FNB strata was then combined with the Mantel–Haenszel method, with evidence of an adjusted RR of 1.04 (95% Confidence Interval 1.00–1.01).

## 4. Discussion

In this meta-analysis, we included seven out of nine studies with a total of 1178 patients. The exclusion of the studies by Itonaga et al. [23] and Seicean et al., 2015 [20] from the quantitative analysis was due to the possible high risk of bias related to the study designs. In fact, as previously reported in the Risk Of Bias section, both studies were designed as crossover studies, and this could have led to bias in the conduction of ECEUS-FNA after EUS-FNA. Indeed, the outcome of the first procedure could influence the second: if the first pass in EUS-FNA is successful, the physician could choose the same region for ECEUS-FNA in order to obtain an adequate specimen; on the contrary, if the first pass does not retrieve an adequate sample, the physician could be biased to change the area of sampling. Moreover, Itonaga et al. [23] reported a different outcome definition, that is, adequate sampling rate instead of established histopathological diagnosis. Although the authors defined sensitivity and specificity, they did not publish the histological diagnosis in ECEUS and EUS groups, and therefore, it was not possible to compare the proportion of established diagnosis. 

Since one study included both pancreatic lesions and retroperitoneal lymph nodes [22], we only considered the subanalysis of patients with pancreatic lesions.

The quantitative analysis of the seven selected studies showed a slight trend of more diagnoses and the greater efficacy of a single pass in patients who undergo CEUS. However, the data currently available do not show statistically significant differences between the two methods (ECEUS vs. EUS). The occurrence of periprocedural adverse reactions in the ECEUS group was substantially comparable to that documented in the standard EUS group. 

These results are clashing with those of a recent meta-analysis [12], published in 2021, which instead concluded that ECEUS-FNA seemed to be superior to EUS-FNA for the diagnosis of pancreatic masses. This discordance may be due to the lower number of studies and patients included in the previous meta-analysis [12] and the different outcomes explored. Indeed, since the outcomes are quite heterogenous between all studies included, we focused on the data that were available and comparable across most of the studies. Moreover, we excluded two studies [20,23] that were previously included in the meta-analysis of Facciorusso et al. [12] and that we judged at high risk of bias according to the ROBINS-I tool.

Our study has several limitations, mainly due to the small number of studies and patients included. Although all studies were homogenous in terms of procedure, they were heterogenous in their outcomes. Indeed, although eight studies explored the number of correct histological diagnoses as the main outcome, one study focused on the adequate sampling rate as the main outcome [23]. Furthermore, the optimal number of needle passes and periprocedural adverse events were not described by each study. Despite the described limitations, the routine application of ECEUS does not seem to improve the efficacy of pancreatic solid lesion sampling. 

## 5. Conclusions

Neither ECEUS-FNA nor FNB appears to be superior to standard EUS-FNA or FNB for the diagnosis of solid pancreatic lesions. These data need to be further explored in future studies.

## Figures and Tables

**Figure 1 cancers-16-01658-f001:**
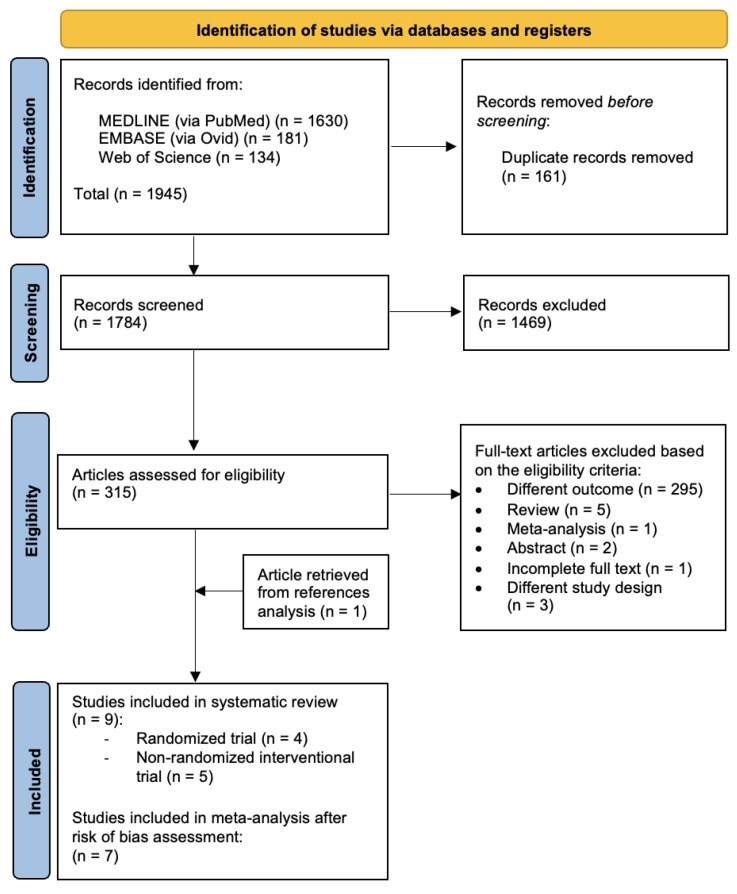
PRISMA study selection flow diagram. PRISMA: preferred reporting items for systematic reviews and meta-analyses.

**Figure 2 cancers-16-01658-f002:**
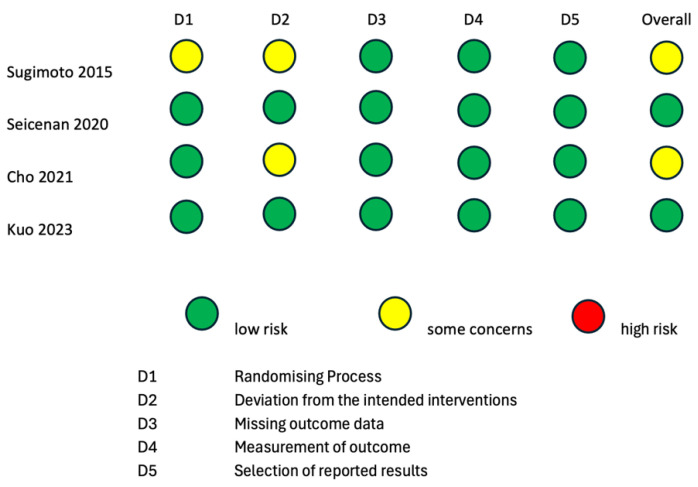
Risk-of-bias assessment according to RoB2 tool for quality assessment of randomized studies. RoB2: a revised Cochrane risk-of-bias tool for randomized trials [15,16,17,18].

**Figure 3 cancers-16-01658-f003:**
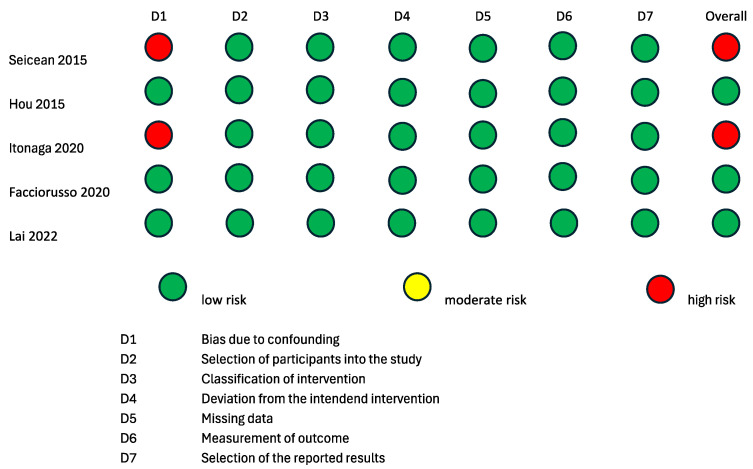
Risk of bias assessment according to ROBINS-I tool for quality assessment of non-randomized interventional studies. ROBINS-I: a tool for assessing risk of bias in non-randomized studies of interventions [17,19,21,22,23].

**Table 1 cancers-16-01658-t001:** Study characteristics of included publications. ECEUS-FNA, endoscopic contrast-enhanced ultrasound fine-needle aspiration; EUS-FNA, endoscopic ultrasound fine-needle aspiration; ECEUS-FNB, endoscopic contrast-enhanced ultrasound fine-needle biopsy; EUS-FNB endoscopic ultrasound fine-needle biopsy; PL, pancreatic lesions; RL, retroperitoneal lymph nodes.

Author, Year	Study Design	Outcomes	Number of Patients	Procedure Type
Sugimoto et al., 2015 [15]	Prospective randomized	correct histological diagnosis optimal number of needle passes required adverse effects	40	ECEUS-FNA (20 patients) - EUS-FNA (20 patients)
Cho et al., 2021 [16]	Prospective randomized	correct histological diagnosis optimal number of needle passes required adverse effects	240	ECEUS-FNA (120 patients) - EUS-FNA (120 patients)
Kuo et al., 2023 [18]	Prospective randomized	correct histological diagnosis optimal number of needle passes required adverse effects	118	ECEUS-FNB (59 patients) - EUS-FNB with fanning technique (59 patients)
Seicean et al., 2020 [17]	Prospective randomized	correct histological diagnosis	148	ECEUS-FNA (74 patients) - EUS-FNA (74 patients)
Hou et al., 2015 [19]	Retrospective cohort study	correct histological diagnosis adequacy of specimen adverse effects	163	ECEUS-FNA (58 patients) - EUS-FNA (105 patients)
Itonaga et al., 2020 [23]	Prospective cohort study	adequate sampling rate adverse effects	93	ECEUS-FNA (93 patients) - EUS-FNA (93 patients)
Seicean et al., 2015 [20]	Prospective cohort study	correct histological diagnosis adverse effects	51	ECEUS-FNA (51 patients) - EUS-FNA (51 patients)
Facciorusso et al., 2020 [21]	Prospective cohort study	correct histological diagnosis adequacy of specimen adverse effects	206	ECEUS-FNA (103 patients) - EUS-FNA (103 patients)
Lai et al., 2022 [22]	Retrospective cohort study	correct histological diagnosis optimal number of needle passes required	133 (115 PL/18 RL)	ECEUS-FNB (40 PL-8 RL) - EUS-FNB (75 PL-10 RL)

**Table 2 cancers-16-01658-t002:** Baseline patients’ characteristics in the included publications. SD, standard deviation (where available); ECEUS, endoscopic contrast-enhanced ultrasound; EUS, endoscopic ultrasound.

Author, Year	Age (Years, SD)	Sex (Male, %)	Study Population
Sugimoto et al., 2015 [15]	68.53 ± 10.2	15 (37.5%)	Solid pancreatic lesions
Cho et al., 2021 [16]	67.30 ± 11.8	127 (52.9%)	Solid pancreatic lesions
Kuo et al., 2023 [18]	64.4 ± 12.1	72 (61%)	Solid pancreatic lesions
Seicean et al., 2020 [17]	64.5 ± 11.3	84 (56.8%)	Solid pancreatic lesions
Hou et al., 2015 [19]	55.65 ± 12.1	99 (60.7%)	Solid pancreatic lesions
Itonaga et al., 2020 [23]	61.4 ± 27.6	50 (53.8%)	Solid pancreatic lesions
Seicean et al., 2015 [20]	61 ± 22	40 (68.9%)	Solid pancreatic lesions
Facciorusso et al., 2020 [21]	66 ± 6 ECEUS 66 ± 8 EUS	113 (54.8%)	Solid pancreatic lesions
Lai et al., 2022 [22]	63.64 ± 12.58	72 (46.5%)	Solid pancreatic lesions and retroperitoneal lymph nodes

**Table 3 cancers-16-01658-t003:** Distribution of histological diagnoses and adverse events among each group.

Author, Year	N° of Established Diagnoses		N° of Adverse Events	
	ECEUS	EUS	ECEUS	EUS
Sugimoto et al., 2015 [15]	20/20 (100%)	20/20 (100%)	0/20 (0%)	0/20 (0%)
Cho et al., 2021 [16]	120/120 (100%)	120/120 (100%)	3/120 (2.5%)	3/120 (2.5%)
Kuo et al., 2023 [18]	59/59 (100%)	59/59 (100%)	1/59 (1.7%)	1/59 (1.7%)
Seicean et al., 2020 [17]	113/148 (76.4%)	112/148 (75.7%)	NR	NR
Hou et al., 2015 [19]	56/58 (96.7%)	91/105 (86.7%)	0/58 (0%)	0/105 (0%)
Itonaga et al., 2020 [23]	NR	NR	1/93 (1.1%)	1/93 (1.1%)
Seicean et al., 2015 [20]	45/51 (88.2%)	41/51 (80.4%)	0/51 (0%)	0/51 (0%)
Facciorusso et al., 2020 [21]	92/103 (89.3%)	85/103 (82.5%)	0/103 (0%)	0/103 (0%)
Lai et al., 2022 [22]	38/40 (95%)	69/75 (92%)	NR	NR

**Table 4 cancers-16-01658-t004:** Cumulative diagnostic accuracy with each needle pass among each group.

Author, Year	Cumulative Diagnostic Accuracy with Each Needle Pass, ECEUS-FNA/FNB		Cumulative Diagnostic Accuracy with Each Needle Pass, EUS-FNA/FNB	
	Needle Pass	Accuracy (%)	Needle Pass	Accuracy (%)
Sugimoto et al., 2015 [15]	1	12/20 (60%)	1	5/20 (25%)
	≤2	15/20 (75%)	≤2	13/20 (65%)
	≤3	18/20 (90%)	≤3	19/20 (95%)
	≤4	20/20 (100%)	≤4	19/20 (95%)
	≤5	20/20 (100%)	≤5	20/20 (100%)
Cho et al., 2021 [16]	1	84/120 (70%)	1	80/120 (66.7%)
	≤2	96/120 (80%)	≤2	100/120 (83.3%)
	≤3	102/120 (85%)	≤3	106/120 (88.3%)
	≤4	103/120 (85.8%)	≤4	106/120 (88.3%)
	≤5	103/120 (85.8%)	≤5	106/120 (88.3%)
Kuo et al., 2023 [18]	1	45/59 (76.3%)	1	45/59 (72.9%)
	≤2	54/59 (91.5%)	≤2	54/59 (91.5%)
	≤3	55/59 (93.2%)	≤3	55/59 (93.2%)
	≤4	57/59 (93.2%)	≤4	57/59 (96.6%)
	≤5	57/59 (96.6%)	≤5	57/59 (96.6%)
	≤6	58/59 (98.3%)	≤6	58/59 (98.3%)

**Table 5 cancers-16-01658-t005:** Distribution of correct diagnosis between patients undergoing EUS-FNA and ECEUS-FNA with relative χ² test. ECEUS-FNA, endoscopic contrast-enhanced ultrasound fine-needle aspiration; EUS-FNA, endoscopic ultrasound fine-needle aspiration.

	Inconclusive Diagnosis, n (%)	Established Diagnosis, n (%)	Total, n
EUS-FNA	74 (11.8)	556 (88.3)	630
ECEUS-FNA	50 (9.1)	498 (90.9)	548
Total	124 (10.5)	1054 (89.5)	1178

*p* = 0.14.

**Table 6 cancers-16-01658-t006:** Distribution of correct diagnosis at first needle pass between patients undergoing EUS-FNA and ECEUS-FNA, with relative χ² test. ECEUS-FNA, endoscopic contrast-enhanced ultrasound fine-needle aspiration; EUS-FNA, endoscopic ultrasound fine-needle aspiration.

	Inconclusive Diagnosis at First Needle Pass, n (%)	Established Diagnosis at First Needle Pass, n (%)	Total, n
EUS-FNA	69 (34.7)	130 (65.3)	199
ECEUS-FNA	58 (29.2)	141 (70.9)	199
Total	127 (31.9)	271 (68.1)	398

*p* = 0.24.

**Table 7 cancers-16-01658-t007:** Cumulative incidence of adverse events between patients undergoing EUS-FNA and ECEUS-FNA, with relative χ² test. ECEUS-FNA, endoscopic contrast-enhanced ultrasound fine-needle aspiration; EUS-FNA, endoscopic ultrasound fine-needle aspiration.

	EUS-FNA/FNB (n = 551)	ECEUS-FNA/FNB (n = 504)	Total (n = 849)	*p*
Adverse events, n (%)	5 (0.9)	5 (1.0)	10 (1.0)	0.86

*p* = 0.89.

## Data Availability

The raw data supporting the conclusions of this article will be made available by the authors on request.

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
