# Peer review of "Endoscopic Contrast-Enhanced Ultrasound and Fine-Needle Aspiration or Biopsy for the Diagnosis of Pancreatic Solid Lesions: A Systematic Review and Meta-Analysis"

_cancers, 2024, doi:10.3390/cancers16091658_

Round 1
Reviewer 1 Report
Comments and Suggestions for Authors
Review
Endoscopic Contrast Enhanced Ultrasound and Fine Needle Aspiration for diagnosis of pancreatic solid lesions: a Systematic Review and Meta-analysis of Individual Participant Data.
I would like to thank the authors for putting in the effort of performing an extensive systematic review on this topic. The review is not only about the comparison between ECEUS-FNA and EUS-FNA, how the title suggests, but also includes FNB. A comprehensive literature search was performed to explore if ECEUS-FNA/FNB could increase diagnostic accuracy for pancreatic lesions and reduce the number of needle passes and adverse effects compared to standard EUS-FNA/FNB. Nine studies were included in the systematic review and after risk assessment, seven studies remained for the meta-analysis including 1178 patients. There were heterogeneous outcome and 2 studies included FNB. ECEUS is not superior to standard EUS-FNA/FNB for diagnosis of solid pancreatic lesions, nor did it affect number of passes or adverse effects.
This systematic review and meta-analysis is different from the conclusion drawn in the recently published meta-analysis by Facciorusso et al (2021). It includes three new studies of which two included FNB instead of FNA. The previous meta-analysis only included FNA-studies. In that respect, this meta-analysis adds to the known literature and summarizes what is known so far. However, I am not sure whether the study by Kuo et al (2023) should be included since it compares ECEUS-FNB with standard EUS-FNA with fanning technique. This is a whole different comparison.
Despite the fact that it may be interesting and relevant for the field, there are a few general and specific concerns. The manuscript is presented in a well-structured manner, however, some things are confusing. For example, it is often not clear from the text, or even the title, that the authors compared ECEUS-FNA or FNB with standard EUS-FNA or FNB. It often seems as if only FNA was taken into account. Please adjust throughout the manuscript and in the title. Also, it seems as if the authors focused their results on solid pancreatic lesions, but then they also included other pancreatic lesions and retroperitoneal lymph nodes in one study. If you did not analyze it and it was described separately, then I suggest to leave it out since it may be confusing to the reader.
Another point of interest is the expected or potential difference between FNA and FNB and the added value of ECEUS in that respect. We know by now that FNB (newer generation needles) outperform FNA. Therefore, it seems debatable whether ECEUS has any additional effect when using the newest generation FNB needles. This point needs to be addressed.
Please carefully check your references since most references from your introduction are from >5-10 years ago.
Introduction
Lines 67-69: ‘Moreover – effects’. Why is this relevant? Please further explain. And moreover, it now seems as if the authors are going to look at the difference between fna and fnb, but in the rest of the article, I cannot find this comparison.
2.0 Materials and Methods
Line 77: ‘the diagnostic success’; please specify how you defined this.
Line 92: I cannot find Table S1.
2.7 Outcomes
See my previous comment on FNA and FNB. A suggestion is to refer to ‘EUS-FNA and/or FNB’ as ‘EUS-guided tissue acquisition’ (EUS-TA). It is then much clearer to the reader if you specifically mean FNA or FNB if you then write FNA or FNB separately.
3. Results
Line 163; ‘all nine’ should be ‘seven’, according to your Figure 1.
Table 2: I prefer only male or female sex with a percentage. Age (years, with standard deviation??)
What type of pancreatic lesions did Facciorusso et al, 2020 include? What percentage was solid? This may influence the results and comparability to other studies included.
Table 3: Suggestion to add percentages.
Title: I also thought that histological diagnosis is used for FNB and cytopathological diagnosis for FNA?
3.4 Please add which papers the individual patient data are derived from.
Table 5 and 6: This seems to be the other way around. The numbers at established diagnosis are mixed up with the numbers from inconclusive diagnosis it seems.
Table 7; can you elaborate on what sort of adverse events? Severity?
Author Response
I would like to thank the authors for putting in the effort of performing an extensive systematic review on this topic. The review is not only about the comparison between ECEUS-FNA and EUS-FNA, how the title suggests, but also includes FNB. A comprehensive literature search was performed to explore if ECEUS-FNA/FNB could increase diagnostic accuracy for pancreatic lesions and reduce the number of needle passes and adverse effects compared to standard EUS-FNA/FNB. Nine studies were included in the systematic review and after risk assessment, seven studies remained for the meta-analysis including 1178 patients. There were heterogeneous outcome and 2 studies included FNB. ECEUS is not superior to standard EUS-FNA/FNB for diagnosis of solid pancreatic lesions, nor did it affect number of passes or adverse effects.
Comment: This systematic review and meta-analysis is different from the conclusion drawn in the recently published meta-analysis by Facciorusso et al (2021). It includes three new studies of which two included FNB instead of FNA. The previous meta-analysis only included FNA-studies. In that respect, this meta-analysis adds to the known literature and summarizes what is known so far. However, I am not sure whether the study by Kuo et al (2023) should be included since it compares ECEUS-FNB with standard EUS-FNA with fanning technique. This is a whole different comparison.
Response: Thanks for your observation. We noticed that we made a mistake in the tables and in the text, describing it as ECEUS-FNB vs EUS-FNA, where instead Kuo et al (2023) compared ECEUS-FNB with standard EUS-FNB. Sorry for that mistake, we made the adjustments along the entire paper.
Despite the fact that it may be interesting and relevant for the field, there are a few general and specific concerns. The manuscript is presented in a well-structured manner, however, some things are confusing. For example, it is often not clear from the text, or even the title, that the authors compared ECEUS-FNA or FNB with standard EUS-FNA or FNB. It often seems as if only FNA was taken into account. Please adjust throughout the manuscript and in the title. Also, it seems as if the authors focused their results on solid pancreatic lesions, but then they also included other pancreatic lesions and retroperitoneal lymph nodes in one study. If you did not analyze it and it was described separately, then I suggest to leave it out since it may be confusing to the reader.
Response: Thank you for your comments. We modified the text and the title to better point out that our analysis is both on FNA and FNB. We included only solid pancreatic lesions. Facciorusso et al. 2020 in Table 2 was incorrectly reported as “Pancreatic Lesions” whereas the authors included only Solid pancreatic lesions. Regarding Lai et al. 2022, the authors included both Pancreatic Lesions and Retroperitoneal Lymphnodes, but they provided separate data (EUS vs ECEUS) for Solid Pancreatic Lesions (adenocarcinoma and neuroendocrine tumors) and Retroperitoneal Lymphnodes. Therefore, we only included Solid Pancreatic Lesions in our analysis.
Another point of interest is the expected or potential difference between FNA and FNB and the added value of ECEUS in that respect. We know by now that FNB (newer generation needles) outperform FNA. Therefore, it seems debatable whether ECEUS has any additional effect when using the newest generation FNB needles. This point needs to be addressed.
Response: Thank you for the observation. We managed to include a subanalysis on the role of ECEUS in FBN compared to ECEUS in FNA. As expected ECEUS does not seem to have additional effect when using FNB. You can find the subanalysis in section “3.4 ECEUS vs standard FNA or biopsy of pancreatic solid lesions: meta-analysis”.
Comment: Please carefully check your references since most references from your introduction are from >5-10 years ago.
Response: We updated the references from the introduction. Thanks for the tip.
Introduction
Comment: Lines 67-69: ‘Moreover – effects’. Why is this relevant? Please further explain.
Response: The number of needle passes can vary between standard EUS-FNA/FNB and ECEUS-FNA/FNB since ECEUS could help the physician in the choice of a better area for sampling, therefore increasing diagnostic accuracy and reducing adverse events related to multiple needle passes. We tried to verify if this difference exists, analyzing the proportion of diagnosis at first needle pass.
Comment: And moreover, it now seems as if the authors are going to look at the difference between fna and fnb, but in the rest of the article, I cannot find this comparison.
Response: As already suggested, we managed to include a subanalysis between FNA and FNB with regards to ECEUS application. Thank you for your observation.
2.0 Materials and Methods
Comment: Line 77: ‘the diagnostic success’; please specify how you defined this.
Response: We define “diagnostic success” the capability to obtain the correct histopathological or cytopathological diagnosis. We modified the text in the “2.7 Outcome” section in order to clarify this detail.
Comment: Line 92: I cannot find Table S1.
Response: I’m sorry, there has been an error in uploading the file with Supplementary Materials. We provided Table S1.
2.7 Outcomes
Comment: See my previous comment on FNA and FNB. A suggestion is to refer to ‘EUS-FNA and/or FNB’ as ‘EUS-guided tissue acquisition’ (EUS-TA). It is then much clearer to the reader if you specifically mean FNA or FNB if you then write FNA or FNB separately.
Response: Thanks again for this tip. We modified the entire text accordingly.
Results
Comment: Line 163; ‘all nine’ should be ‘seven’, according to your Figure 1.
Response: We modified the text. We described all nine studies in the qualitative synthesis, but after the Risk of Bias assessment we included only seven studies in the quantitative analysis.
Comment: Table 2: I prefer only male or female sex with a percentage. Age (years, with standard deviation??)
Response: We modified the table accordingly. Age comes with standard deviation, if available in the paper.
Comment: What type of pancreatic lesions did Facciorusso et al, 2020 include? What percentage was solid? This may influence the results and comparability to other studies included.
Response: They included only solid pancreatic lesions as specified in their abstract. We modified Table 2 and the text accordingly.
Comment: Table 3: Suggestion to add percentages.
Response: We added percentages as suggested.
Comment: Title: I also thought that histological diagnosis is used for FNB and cytopathological diagnosis for FNA?
Response: Yes. The outcome was to compare diagnostic success, intended as the number of correct histological or cytopathological diagnosis between EUS-guided sampling and ECEUS-guided sampling.
Comment: 3.4 Please add which papers the individual patient data are derived from.
Response: Unfortunately, there has been and error in identifying the type of meta-analysis. We didn’t have access to databases and single patient’s data. Our quantitative analysis is purely based on data retrievable from the published papers and therefore is a standard meta-analysis and not an individual patient data meta-analysis. This error has been pointed out by the second reviewer and we modified the title, text and references.
Comment: Table 5 and 6: This seems to be the other way around. The numbers at established diagnosis are mixed up with the numbers from inconclusive diagnosis it seems.
Response: Thanks for your observation. We corrected the tables.
Table 7; can you elaborate on what sort of adverse events? Severity?
Response: The adverse events reported by the authors were mainly post-procedural bleedings and, in some minor cases, acute pancreatitis. We added a statement in the “Outcomes” section to clarify this detail.
Reviewer 2 Report
Comments and Suggestions for Authors
Individual patient meta-analyses represent a huge work and a very valuable source of data; however, this is NOT an individual patient meta-analysis as the authors did not have access to the datasets of the single studies. This is a standard meta-analysis on an important topic, so still interesting but not with the same value of IPD SRMA. The authors should change the title and correct the text accordingly.
Was the availability of ROSE considered in the analysis?
Was the proportion of surgical patients the same in the included studies?
Author Response
Comment: Individual patient meta-analyses represent a huge work and a very valuable source of data; however, this is NOT an individual patient meta-analysis as the authors did not have access to the datasets of the single studies. This is a standard meta-analysis on an important topic, so still interesting but not with the same value of IPD SRMA. The authors should change the title and correct the text accordingly.
Response: We are sorry to have mistaken IPD MA with standard meta-analysis. We managed to correct the title, text and references accordingly. Thanks for this valuable and important comment on the nature of the paper. We are truly sorry for this mistake.
Comment: Was the availability of ROSE considered in the analysis?
Response: Only 2 papers (Kuo et al and Sugimoto et al 2015) used ROSE as preliminary evaluation and therefore we didn’t take it into account.
Comment: Was the proportion of surgical patients the same in the included studies?
Response: Thank for the interesting question. All the included studies considered both surgical and non-surgical patients, but only one included study reported in terms of percentage the proportion of these two groups (Hou et al 35%). Therefore it was not possible to compare the proportion of final diagnosis obtained via surgery across all the included studies.
Round 2
Reviewer 1 Report
Comments and Suggestions for Authors
Dear authors,
Thank you for the comprehensive and clearly written response. I am satisfied with your answers and adjustments accordingly. The only remaining suggestion I have is to change your conclusion and leave the 'trend-sentence' out of it. It is confusing since a trend does not say much and it only applies to FNA and not to FNB.
Author Response
As suggested we modified the conclusions (both in the abstract and main text), removing the "slight trend" sentence. We agree with your observation.
Thanks for your insight and your revision work.
Reviewer 2 Report
Comments and Suggestions for Authors
The revised version of the manuscript is OK. Thank you!
Author Response
Thanks for your work and your insight upon this paper.